# TESTING TYPICALITY WITH RESPECT TO AN ENSEMBLE OF LEARNED DISTRIBUTIONS

## ABSTRACT

Good methods of performing anomaly detection on high-dimensional data sets are needed, since algorithms which are trained on data are only expected to perform well on data that is similar to the training data. There are theoretical results on the ability to detect if a population of data is likely to come from a known base distribution, which is known as the goodness-of-fit problem, but those results require knowing a model of the base distribution. The ability to correctly reject anomalous data hinges on the accuracy of the model of the base distribution. For high dimensional data, learning an accurate-enough model of the base distribution such that anomaly detection works reliably is very challenging, as many researchers have noted in recent years. Existing methods for the goodness-of-fit problem do not account for the fact that a model of the base distribution is learned. To address that gap, we offer a theoretically motivated approach to account for the density learning procedure. In particular, we propose training an ensemble of density models, considering data to be anomalous if the data is anomalous with respect to any member of the ensemble. We provide a theoretical justification for this approach, proving first that a test on typicality is a valid approach to the goodness-of-fit problem, and then proving that for a correctly constructed ensemble of models, the intersection of typical sets of the models lies in the interior of the typical set of the base distribution. We present our method in the context of an example on synthetic data in which the effects we consider can easily be seen.

## 1 INTRODUCTION

Machine learning models are inherently non-robust to distributional shift, and at no fault of the model necessarily. There is no reason to expect that models should perform well on data that is dissimilar to the data on which they were trained. Interestingly, despite the fact that researchers and practitioners have been able to train models that perform exceptionally well on a variety of challenging tasks, we are still bad at reliably predicting when those models will fail. This implies that not only will the models have undefined behavior on out-of-distribution data, we are unable to detect when the models are presented with out-of-distribution data. This poses a conundrum, since we wish to deploy our high-performing models, yet often we can't since they can potentially have unpredictable behavior at unpredictable instances.

Since training a model that is robust to all possible distributional shifts the model might encounter is potentially impossible, a more modest approach might be to come up with ways for detecting out-of-distribution data. These detections could then act as an indication that the model might be incorrect. This ability to predict incorrectness can go a long way in making systems more reliable.

The problem of detecting out-of-distribution data has a long history, and is formally known as the goodness-of-fit problem. Statisticians have proved bounds on the probability of detecting populations of out-of-distribution data, such as in Barron (1989) and Balakrishnan et al. (2019). These type of bounds show that certain tests can be performed which are capable of discerning (with non-trivial probability) that populations of data sampled from distributions at least some positive distance away from the base distribution are anomalous. However, in order to perform the proposed tests, an explicit form of the probability density function (or probability mass function) describing the base distribution is needed. For most real-world data sets, this density is not known, and must be esti-

mated. While there has been a lot of analysis on the ability to detect anomalous data, those analyses typically do not account for the fact that the base density for which the tests are designed is learned.

Empirically, however, many researchers have noted that detecting anomalous data in high dimensions using learned densities is hard, even when using modern, powerful density estimators. For example, the researchers in Nalisnick et al. (2018) and Choi & Jang (2018) both claim that state-of-the-art learned densities are not suitable for anomaly detection since they assign higher probability to some out-of-distribution data than the data on which the models were trained. The authors in Nalisnick et al. (2019) realized that such a phenomenon is actually not cause for alarm, and is even expected with high-dimensional distributions. Therefore they propose performing an anomaly detection test based on the typicality of data under the learned density instead of the likelihood of data under the learned density. However, in that work, the authors noted such a test still performed poorly in some cases.

In this article, we investigate why goodness-of-fit tests are challenging when using learned densities. In particular, we analyze how the typical set of learned distributions relates to the typical set of the ground truth distribution. In order to do this, we work with synthetic distributions in which the probability density function is known exactly. The contributions we make are summarized as the following:

- We prove error rate bounds on the goodness-of-fit test based on testing for typicality with respect to the base distribution

- We prove theorems stating that the typical sets of any two distributions having sufficiently high KL divergence must have low intersection, and that distributions having low KL divergence must have non-zero intersection.

- We use these theorems to motivate our proposed method for conservative goodness-of-fit testing. Specifically, we propose training an ensemble of models, and show that by taking the intersection of their typical sets, we can approximately recover the typical set of the ground truth distribution. We show that such an ensemble can exist and give sufficient conditions for its existence.

- We demonstrate on synthetic data sets that the typical set of standard learned distributions and the ground-truth distribution often have low intersection, even when the class of densities from which we approximate the ground truth contains the ground-truth. We validate that our proposed method addresses this issue.

## 2 PRELIMINARIES

The setting we are concerned with is the following. Consider a continuous random variable $x \in \mathcal{X}$ with probability density function $p(x)$. Let $x^n := (x_1, x_2, ..., x_n)$ denote the collection of $n$ random variables $x$. Let $h_p$ denote the differential entropy of $p(x)$. For $\epsilon > 0$ and positive integer $n$, The typical set with respect to $p(x)$ is defined as the following:

$$T_\epsilon^{(n)} := \left\{ x^n \in \mathcal{X}^n : | -\frac{1}{n} \log p(x^n) - h_p | < \epsilon \right\}. \tag{1}$$

In words, this is the set of sequences of samples whose average negative-log-probability is close to the differential entropy. This set has high probability under $p(x)$, implying that most sequences of data points sampled from $p(x)$ are contained in this set (Cover & Thomas, 2012).

### 2.1 HYPOTHESIS TESTING

We are interested in determining the plausibility that some collection of samples $\tilde{x}^n \in \mathcal{X}^n$ were sampled i.i.d. according to $p(x)$. In hypothesis testing terminology, we are interested in determining which of two hypotheses are more probable:

- The *null-hypothesis*,  $\quad\quad\quad\quad \mathcal{H}_0 := \tilde{x}^n \sim p(x)$
- The *alternative-hypothesis*,  $\quad\quad \mathcal{H}_1 := \tilde{x}^n \sim \hat{p}(x),$

where here $\hat{p}(\mathrm{x})$ is any distribution such that $D_{\mathrm{KL}}(p\|\hat{p}) \geq d$. If we define $A_n \subset \mathcal{X}^n$ to be the acceptance region for the null-hypothesis, or the set of all sequences $\mathrm{x}^n$ such that $\mathcal{H}_0$ is deemed more probable, then we can define the probabilities of error as the following:

$$\alpha_n = p^n(A_n^{\mathsf{c}}), \quad \beta_n = \max_{\tilde{p}:D(p\|\tilde{p})\geq d} \tilde{p}^n(A_n). \tag{2}$$

Typically we desire tests that minimize $\beta_n$ for a fixed $\alpha_n$, or tests that minimize some linear combination of the two.

The authors in Nalisnick et al. (2019) propose a test procedure which accepts the null-hypothesis iff $\tilde{\mathrm{x}}^n \in T_\epsilon^{(n)}$. In other words, a collection of points are determined to be anomalous with respect to $p(\mathrm{x})$ iff $\tilde{\mathrm{x}}^n \notin T_\epsilon^{(n)}$. In that work, they show that this test performs about as well as other tests such as the Student's t-test, the Kolmogorov-Smirnov test, the Maximum Mean Discrepancy test, and the Kernelized Stein Discrepancy test. We note that in those comparisons, all tests are performed with respect to a learned density as a proxy for the ground-truth density.

In this work, we consider the same typicality-based test for accepting or rejecting the null hypothesis. We focus on this test since it allows for analysis into what happens when the proxy learned density for which the typical set is constructed is different from the ground-truth distribution $p(\mathrm{x})$. In order to validate this approach, we first prove that for a fixed $\alpha_n$, the typical set test achieves an error rate $\beta_n$ that is at worst, given by:

**Theorem 1.** *Let the region of acceptance for the typicality test be defined by the set $T_\epsilon^{(n)}(p(\mathrm{x}))$. For $n$ sufficiently large, then $\alpha_n < \epsilon$, and*

$$\beta_n < \max_{\tilde{p}:D(p\|\tilde{p})\geq d} e^{-n(D_{\mathrm{KL}}(\tilde{p}\|p)+h_{\tilde{p}}-h_p-3\epsilon)} + 3\epsilon$$

*Proof of Theorem 1.* See Section A.1. $\qquad\square$

This theorem states that for a fixed rate of correctly accepting $\mathcal{H}_0$, the test for typicality will fail to detect anomalous data at a rate no greater than the bound given.

## 2.2 DENSITY LEARNING AS OPTIMIZATION

In order to perform the test on typicality to determine whether a sequence of data points are anomalous or not, a model of the probability density function $p(\mathrm{x})$ is needed. For many practical applications, we are only ever presented with a data set of $N$ samples from $p(\mathrm{x})$, which we denote $\bar{X} = (\bar{\mathrm{x}}_1, \bar{\mathrm{x}}_2, ...\bar{\mathrm{x}}_N)$. However, an estimate of the ground-truth density can be learned. We denote a parameterized distribution by $q(\mathrm{x}; \boldsymbol{\theta})$, which can be learned by minimizing the following objective:

$$\min_{\boldsymbol{\theta}} D_{\mathrm{KL}}(p(\mathrm{x})\|q(\mathrm{x}; \boldsymbol{\theta})). \tag{3}$$

The KL divergence is defined by $D_{\mathrm{KL}}(p(\mathrm{x})\|q(\mathrm{x}; \theta)) := \mathbb{E}_p[log \frac{p(\mathrm{x})}{q(\mathrm{x};\theta)}]$. By expanding terms and eliminating constants, it is clear that an equivalent optimization problem is

$$\max_{\boldsymbol{\theta}} \mathbb{E}_p[log\ q(\mathrm{x}; \boldsymbol{\theta})] \approx \max_{\boldsymbol{\theta}} \frac{1}{N} \sum_{i=1}^{N} log\ q(\bar{\mathrm{x}}_i; \boldsymbol{\theta}). \tag{4}$$

The optimal $\boldsymbol{\theta}$ in this problem is that which maximizes the log likelihood on the data set $\bar{X}$. This is why this optimization is commonly referred to as maximum-likelihood estimation. Importantly, the dependence on the unknown true distribution $p(\mathrm{x})$ was removed through the use of the law of large numbers. Since the KL-divergence is equal to zero if and only if $p(\mathrm{x}) = q(\mathrm{x}; \boldsymbol{\theta}) \ \forall x \in \mathcal{X}$, this objective is well-motivated if we wish that $q(\mathrm{x}; \boldsymbol{\theta}) \approx p(\mathrm{x})$. However, there are many other statistical divergences between distributions that would also be suitable as objectives in the optimizations above. In fact, many of these other divergences might be preferred, especially in the context of anomaly detection. Among other reasons, this is because when the parameterized class of distributions $q(\mathrm{x}; \boldsymbol{\theta})$ is highly expressive, the optimizations are non-convex, and local optima may be found with sub-optimal properties. This can be true even if there exist some $\boldsymbol{\theta}^*$ such that $q(\mathrm{x}; \boldsymbol{\theta}^*) = p(\mathrm{x})$.

For example, by looking at the form of the KL-divergence, there is no direct penalty for $q(x; \boldsymbol{\theta})$ in assigning a high density to points far away from any of the $\bar{x}_i$'s. This can lead to locally optimal learned densities which have high probability density in regions of $\mathcal{X}$ that $p(x)$ does not. This can potentially lead to situations in which a collection of points can be anomalous with respect to $p(x)$, but in-distribution with respect to the sub-optimally learned $q(x; \boldsymbol{\theta})$.

There exist other divergences which do not suffer this effect, but it is not obvious how to optimize with respect to them. For example, the *reverse* KL-divergence, which switches the places of $p(x)$ and $q(x; \boldsymbol{\theta})$ in the standard KL-divergence, has an alternate effect; $q(x; \boldsymbol{\theta})$ is directly penalized for assigning high density to anywhere that does not also have high density under $p(x)$. Unfortunately, this requires direct knowledge of $p(x)$ to evaluate the objective, as do all divergences other than the forward KL, to the best of our knowledge. This makes the forward KL-divergence special in that it is the only divergence which can directly be optimized for.

These implications are important when considering how learning a density affects the hypothesis testing problem.

## 2.3 MODERN DENSITY PARAMETERIZATIONS

There have been many advancements in the parameterization of density models in recent years. Broadly speaking, there are two main classes of parameterizations, being latent-variable models and invertible flow models. Latent variable models, such as Variational Auto-encoders (Rezende et al., 2014; Kingma & Welling, 2013), refer to models which map a lower-dimensional, latent random variable through a probabilistic mapping into data space. Because the image of any non-surjective function necessarily has measure zero, these type of models cannot have deterministic mappings, or else the model cannot constitute a valid probability distribution. The probabilistic mapping corrects for this, and the ability to represent data in a lower-dimensional latent space is a nice property of such models.

The other major class of parameterizations are those which map a probability density function through a bijective, continuously differentiable mapping, which we refer to as change-of-variable models. These models include autoregressive models, such as PixelCNN (Oord et al., 2016; Salimans et al., 2017), and invertible flow models, such as NICE (Dinh et al., 2014), RealNVP (Dinh et al., 2016), and GLOW (Kingma & Dhariwal, 2018). Often autoregressive models are considered as a separate class of model than invertible flow models, but we lump them together as they are effectively different implementations of the same concept. All change-of-variable models operate on the ability to express a probability density function through a change of variables. Specifically, if $m(x; \boldsymbol{\theta}) : \mathcal{X} \to \mathcal{Z}$ is a continuously differentiable bijection, and $r(z; \boldsymbol{\theta})$ is a probability density function corresponding to random variable $z \in \mathcal{Z}$, then together $m$ and $r$ implicitly define a probability density function over random variable $x \in \mathcal{X}$:

$$q(x; \boldsymbol{\theta}) = r(z; \boldsymbol{\theta}) \left| \frac{dm(x; \boldsymbol{\theta})}{dx} \right|. \tag{5}$$

So long as the determinant in (5) is easy to evaluate, the probability density $q(x; \boldsymbol{\theta})$ can be evaluated and the parameters $\boldsymbol{\theta}$ can easily be optimized over. In both of these broad class of parameterizations, there have been many clever implementations of these concepts, creating highly expressive density models. In this work, our considerations are tangential and complementary to the advancements in these parameterizations. This is because regardless of the parameterization used, all of these models rely on optimizing the forward KL-divergence in order to learn their parameters. The effects we study in this paper occur even when the parameterization includes the ground-truth density we wish to learn, indicating that advancements in the expressivity of the models are unlikely to fix the undesired effects.

## 3 CASE STUDY ON SYNTHETIC DATA

To understand why hypothesis testing is so difficult when using learned densities, we investigate this occurrence by learning synthetic probability distributions for which the ground-truth $p(x)$ is known. In particular, we consider the problem of learning the density of a high-dimensional mixture of Gaussians.

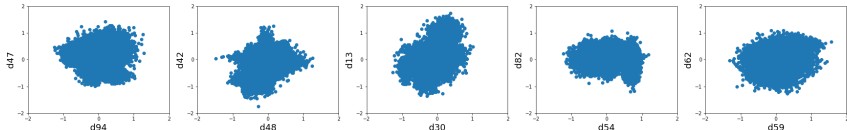

Figure 1: Random projections of samples from a 100-dimensional Mixture of Gaussians Distribution.

Let $M$ denote the number of mixture components in the base distribution. Then we define the base distribution as the following:

$$p(\mathbf{x}) := \sum_{m=1}^{M} \pi_m p_m(\mathbf{x}; \boldsymbol{\mu}_m, \boldsymbol{\Sigma}_m), \qquad (6)$$

where $p_m(\mathbf{x}; \boldsymbol{\mu}_m, \boldsymbol{\Sigma}_m)$ is the probability density function associated with a Gaussian distribution with mean $\boldsymbol{\mu}_m$ and covariance $\boldsymbol{\Sigma}_m$. For this investigation, we define the space $\mathcal{X} := \mathbb{R}^{100}$, so that effects dependent on the dimensionality of $\mathcal{X}$ are evident. Here we consider the parameters $\boldsymbol{\mu}_m$ and $\boldsymbol{\Sigma}_m$ to be chosen such that the $\boldsymbol{\mu}_m$ are sampled from a uniform distribution over a hyper-ball with radius $r = 3.0$, and the matrices $\boldsymbol{\Sigma}_m$ are diagonal matrices with the negative logarithm of the diagonal elements each sampled from a uniform distribution between 1.0 and 3.0. We fix the $\pi_m = 1/M$. In all experiments we choose the number of components to be $M = 20$.

Some random projections of samples from one such $p(\mathbf{x})$ can be seen in Figure 1. Note that although in each projection the probability mass of the different mixture components seem to be overlapping, due to the dimensionality of the space, each mode is separated by a relatively large distance. In the example shown in Figure 1, the minimum distance between the means of each mixture component is 3.62. Furthermore, the max probability of any mode with respect to any of the other mixture components is roughly $1e$-20. This means that each of the $M$ modes in the mixture distribution are clearly distinct and separated by regions of near-zero probability.

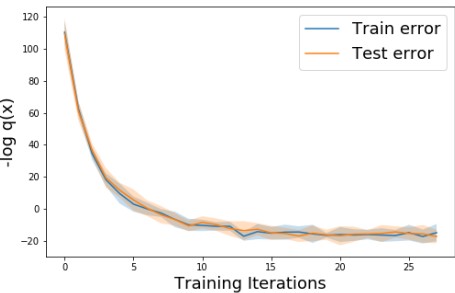

Figure 2: Learning curve for the Mixture of Gaussian Experiment. Average (across experiments) train and test neg-log-likelihoods are plotted along with their $\pm 1$ standard deviation region.

We attempt to learn $p(\mathbf{x})$ by optimizing (4), for a $q(\mathbf{x}; \boldsymbol{\theta})$ parameterized *exactly* as $p(\mathbf{x})$. In other words, the parameters $\boldsymbol{\theta}$ are the set $\{\pi_1, \boldsymbol{\mu}_1, \boldsymbol{\Sigma}_1, \pi_2, \boldsymbol{\mu}_2, \boldsymbol{\Sigma}_2, ..., \pi_M, \boldsymbol{\mu}_M, \boldsymbol{\Sigma}_M\}$, where the $\pi_m > 0$ are constrained to sum to 1, and the $\boldsymbol{\Sigma}_m$ are constrained to be positive definite and diagonal.

We perform five experiments, each time randomly initializing the parameters $\boldsymbol{\theta}$ exactly as was done for $p(\mathbf{x})$. We use the indices $k \in \{1, ..., K = 5\}$ to index the 5 experiments ran, and refer the the $k$-th learned density as $q_k(\mathbf{x}; \boldsymbol{\theta}_k)$. The training curve for learning these densities can be seen in Figure 2, which shows the average test and train errors across experiments, and random projections of the resulting samples overlaid on samples of the base distribution can be seen in Figure 5. Examining these figures, we see that the learning procedure converged, yet the samples generated from the learned distributions clearly differ from the ground-truth samples. By examining the relative probabilities of each mixture component for the learned densities (Figure 3), we see that all of the

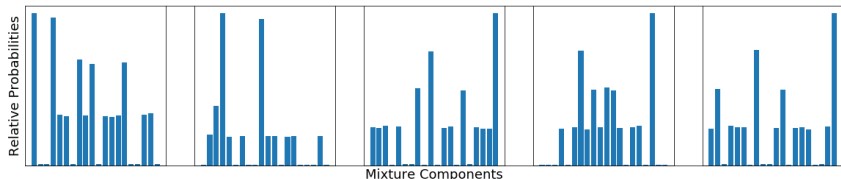

Figure 3: Bar plots showing the relative weighting of the mixture components in the learned densities in the Mixture of Gaussians experiment. Here it can be seen that each of the 5 learned densities use only approximately 12 out of the 20 modes to explain the data.

learned densities attempt to represent the $M = 20$ modes of the base distribution using only about 12 of their 20 modes.

Recalling that the base distribution $p(x)$ has distinctly separated modes, the fact that the base distributions represent the data using fewer modes directly implies that the models necessarily assign high density to regions in-between modes of the ground-truth distribution. This behavior is very troublesome for the purposes of anomaly detection, since intuitively these regions of high-density in-between modes of the true distribution effectively include those regions in the typical set of the learned density. This means that data-points lying in these regions, which should be considered anomalous with respect to the true distribution, are not detectable by hypothesis tests which test for typicality (or likelihood, for that matter).

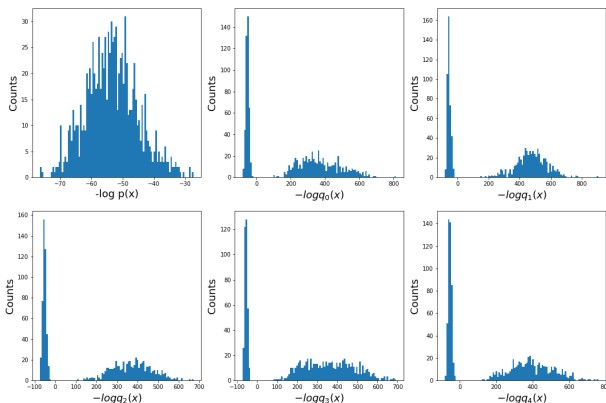

Figure 4: Histograms of the negative log-likelihood of the ground truth distribution, evaluated on samples from the ground-truth distribution (top-left), and samples from each of the learned distributions in the Mixture of Gaussians experiment. As it can be seen, only about a third of samples from each learned distribution have negative log-likelihood close to the ground-truth sample average.

To validate this claim, we generate histograms of the ground-truth negative log-likelihoods on samples from the base distribution and each of the learned distributions, which can be seen in Figure 4. We also list in Table 1 the percentages of samples from each learned density that lie in the typical sets of the ground-truth distribution and every other learned distribution. These statistics give a sense of the intersection of the typical sets of the learned and ground-truth densities. In particular, despite almost all of the ground-truth samples lying in the typical set of each learned-distribution, only about 40% of the samples from each learned distribution lie in the typical set of the true distribution. In the context of anomaly detection, this means that if we used a test for typicality using one of these learned densities, about 60% of the region of acceptance would include points that are not actually typical with respect to the base distribution.

Table 1: Percentages of samples lying in the Typical Set of each of the learned and ground-truth distributions. Here, we take $\epsilon$ to be such that approximately $95\%$ of samples from a distribution lie in its own typical set, and we set $n = 1$. Note that each of the learned densities include most of the ground-truth samples from $p(\mathrm{x})$ in their typical sets, but the typical set of the ground-truth distribution excludes most samples from the learned densities. Furthermore, most of the samples from the learned densities are excluded from the typical sets of other learned densities.

| | Sampling Distribution | | | | | |
|---|---|---|---|---|---|---|
| | $p(\mathrm{x})$ | $q_1(\mathrm{x}; \boldsymbol{\theta}_1)$ | $q_2(\mathrm{x}; \boldsymbol{\theta}_2)$ | $q_3(\mathrm{x}; \boldsymbol{\theta}_3)$ | $q_4(\mathrm{x}; \boldsymbol{\theta}_4)$ | $q_5(\mathrm{x}; \boldsymbol{\theta}_5)$ |
| $T_\epsilon^{(n)}(p(\mathrm{x}))$ | 95.3 | 37.8 | 34.4 | 42.9 | 35.0 | 40.7 |
| $T_\epsilon^{(n)}(q_1(\mathrm{x}; \boldsymbol{\theta}_1))$ | 93.3 | 93.4 | 35.1 | 41.2 | 34.3 | 38.9 |
| $T_\epsilon^{(n)}(q_2(\mathrm{x}; \boldsymbol{\theta}_2))$ | 93.2 | 69.6 | 95.0 | 80.6 | 68.9 | 73.0 |
| $T_\epsilon^{(n)}(q_3(\mathrm{x}; \boldsymbol{\theta}_3))$ | 96.2 | 47.0 | 33.9 | 97.0 | 40.5 | 47.4 |
| $T_\epsilon^{(n)}(q_4(\mathrm{x}; \boldsymbol{\theta}_4))$ | 94.5 | 47.4 | 44.5 | 44.8 | 94.5 | 50.4 |
| $T_\epsilon^{(n)}(q_5(\mathrm{x}; \boldsymbol{\theta}_5))$ | 95.7 | 46.6 | 39.9 | 48.9 | 38.2 | 50.4 |

The mismatch in typical sets corroborates what was stated in Section 2.2, which was that the forward KL-divergence objective can lead to assigning density to regions far from any data points. Here, the learned densities are optimized to explain the data. The way in which they do results in inadvertently explaining other regions of the data space as well.

The example presented here is admittedly a simple one, yet it demonstrates that coming up with more clever parameterizations will not necessarily result in a better learning process. In this example, the learned densities are expressive enough to learn the true distribution, however get stuck in poor local minima due to bad initializations. This is the case in this situation, even though we initialized the learned distributions according to the same distribution over parameters that the true distribution was initialized from. For real data sets, it is unlikely that the parameterized space of densities we learn over can ever represent the density we are trying to estimate, and even if it could, finding a proper initialization might be exceptionally difficult. Therefore, the effects illustrated in this example are expected to occur in more complicated (and interestig) domains, and should be accounted for in those domains as well.

In order to address this issue of over-assignment of density, we turn to ensembles of learned distributions. As can be seen in Table 1, we find that due to the random initializations of each of the learned distributions, the regions in which they over-assign density have low intersection. In the following section, we demonstrate how this property can be leveraged to account for the mismatch in typical sets between learned and target densities.

## 4 TYPICAL SETS OF ENSEMBLES OF LEARNED DENSITIES

We propose an alternative test to discern the two hypotheses described in Section 2.1. First we define what we call the multi-typical set, with respect to an ensemble of distributions $q_k(\mathrm{x})$, $k \in \{1, ..., K\}$. For a given $\epsilon > 0$ and positive integer $n$, this set is defined as the following:

$$T_\epsilon^{(n)}(\{q_1(\mathrm{x}), ..., q_K(\mathrm{x})\}) := \left\{ \mathrm{x}^n \in \mathcal{X}^n : \max_{k \in \{1, ..., K\}} | - \log q_k(\mathrm{x}^n) - h_k| < \epsilon \right\}. \quad (7)$$

Here, $h_k$ refers to the differential entropy of the $k$-th density in the ensemble, $q_k(\mathrm{x})$.

The method we propose is the following:

1. Train an ensemble of parameterized densities, $\{q_1(\mathrm{x}; \boldsymbol{\theta}_1, ..., q_K(\mathrm{x}; \boldsymbol{\theta}_K)\}$, using random initializations and using the maximum likelihood objective (4).
2. When discerning between $\mathcal{H}_0$ and $\mathcal{H}_1$ (Section 2.1), we choose $\mathcal{H}_0$ iff $\tilde{\mathrm{x}}^n \in T_\epsilon^{(n)}(\{q_1(\mathrm{x}; \boldsymbol{\theta}_1), ..., q_K(\mathrm{x}; \boldsymbol{\theta}_K)\})$.

Before justifying this test theoretically, we first demonstrate its utility on the Mixture of Gaussians example in Section 3. By using rejection sampling, we generate samples that lie in the multi-

typical set with respect to the 5 learned densities. Specifically, we generate 1000 samples from each model, and then reject any of the resulting 5000 samples that do not lie in the typical set of every model in the ensemble. The result of this is that $39.9\%$ of the total samples make it through the rejection process, and $94.8\%$ of the remaining samples lie within the typical set of the ground-truth distribution. Furthermore, $96.6\%$ of the ground-truth samples are considered typical with respect to the multi-typical set (7). We emphasize that this sampling procedure is a proxy for estimating the interior of the multi-typical set. The fact that about as many of these samples lie in the ground-truth typical set as samples from the ground-truth distribution itself demonstrates that this set works as a good indicator of typicality with respect to the ground-truth distribution.

Therefore, by leveraging the use of an ensemble of learned densities, we have shown that the short-coming of any individual learned density model can be overcome, at least in this motivating example.

The high-level idea for why this method works is that by sufficiently minimizing the forward KL-divergence between the ground-truth distribution and each of the learned distributions, there is guaranteed to be some intersection of the typical sets of each model in the ensemble, and at least part of that intersection must lie in the typical set of the ground-truth typical set. Furthermore, we prove that if each model in the ensemble is sufficiently different, such that the KL-divergence between each model is large enough, then the intersection between their typical sets must be small. This implies that the intersection of typical sets from sufficiently different models trained to have low KL-divergence with the ground-truth model must lie almost entirely in the interior of the typical set of the ground-truth distribution.

To formalize this argument, we state and prove the following theorems:

**Theorem 2.** *For continuous random variable* $\mathrm{x} \in \mathcal{X}$*, Consider some distribution* $p(\mathrm{x})$ *with differential entropy* $h_p$*, and any collection of distributions* $q_k(\mathrm{x})$*,* $k \in \{1, ..., K\}$*, all having* $D_{\mathrm{KL}}(p \| q_k) = d_k < \infty$*. Let* $T_p := T_\epsilon^{(n)}(p)$ *denote the typical set of* $p$*, and similarly* $T_{q_k} := T_\epsilon^{(n)}(q_k)$ *the typical set of* $q_k$ *for each* $k \in \{1, ..., K\}$*.*

*For* $n$ *sufficiently large, if the following hold,*

- $\frac{1}{4\epsilon} \log \left( \frac{1-2\epsilon}{\frac{K-1}{K}+\epsilon} \right) > n$

- $d_k < \frac{1}{n} \log \left( \frac{1-2\epsilon}{\epsilon} e^{-4n\epsilon} - \frac{K-1}{\epsilon K} \right) + 3\epsilon$

*then*

$$Vol\left(T_p \bigcap_k T_{q_k}\right) > 0.$$

**Theorem 3.** *For continuous random variable* $\mathrm{x} \in \mathcal{X}$*, consider two distributions,* $q_a(\mathrm{x})$ *and* $q_b(\mathrm{x})$*. Denote* $h_a$ *and* $h_b$*, the differential entropy of these distributions, respectively. Similarly, let* $T_a := T_\epsilon^{(n)}(q_a)$ *and* $T_b := T_\epsilon^{(n)}(q_b)$ *represent the typical sets of* $q_a$ *and* $q_b$*. If for some* $0 < r \le 1$*, and* $n$ *sufficiently large,*

$$D_{\mathrm{KL}}(q_a \| q_b) > h_b - h_a - 3\epsilon + \frac{1}{n} \log \left( \frac{1}{r(1-\epsilon)e^{-2n\epsilon} - 2\epsilon} \right),$$

*then,*

$$\frac{Vol\left(T_a \cap T_b\right)}{Vol\left(T_a\right)} < r.$$

Proof of both Theorems is given in Section A.1. The result of Theorem 2 is that we can be sure that if every model in a density of learned distributions is successful in minimizing the KL-divergence with respect to the ground-truth density, then at least part of the multi-typical set must lie in the interior of the typical set $T_\epsilon^{(n)}(p(\mathrm{x}))$. The result of Theorem 3 is that sufficiently different models in an ensemble must not have large intersection. The condition should hold for the $r$ such that the ratio of volumes in Theorem 3 is approximately the same for every pair of models in the ensemble. If the conditions listed in both theorems hold, then the multi-typical set must lie almost entirely in $T_\epsilon^{(n)}(p(\mathrm{x}))$. The result of this is that the multi-typical set, if constructed according to the conditions

given, can be used as a conservative estimate of the typical set $T_\epsilon^{(n)}(p(\mathrm{x}))$. Here we use the word conservative, since the conditions only are sufficient for the multi-typical set to act as an under-approximation of $T_\epsilon^{(n)}(p(\mathrm{x}))$.

The point of proving these theorems is to demonstrate that, in theory, an ensemble of learned models can approximate the typical set of the ground-truth distribution. The bounds given are sufficient conditions, but in practice we find that it is much easier to find an ensemble of models such that the multi-typical set approximates the ground-truth typical set than the bounds require. Therefore, these theorems should be taken as proof that such a procedure is well motivated, and not necessarily a guide for choosing the values of $\epsilon$ and $n$ in practice.

## 5 RELATED WORK

There are many works interested in goodness-of-fit testing for modern high-dimensional data distributions. The works most similar to ours are Nalisnick et al. (2019) and Choi & Jang (2018). The authors in Nalisnick et al. (2019) use a test for typicality, showing empirically that it can perform about as well as other traditional goodness-of-fit tests.

The authors in Choi & Jang (2018) leverage an ensemble of learned density models to obtain better tests for anomaly detection than the other tests using a single learned distribution. Specifically, they leverage the Watanabe-Akaike Information Criterion (Watanabe, 2010) to produce a score which averages the log probabilities across models of the ensemble, and subtracts the variance. Without considering the variance term, taking the average log probability is equivalent to the geometric mean of the probability density functions in the ensemble, which acts as a sort of soft-min over the ensemble. Hence, in a way, that method can be thought of as generating a density that is the point-wise least probable density in the ensemble, where as the method we propose can be thought of as taking the point-wise least typical density in the ensemble. We elaborate more on this in Section A.2.

We also point out that the authors in Choi & Jang (2018) propose a baseline test that resembles a test on typicality. They measure the distance from latent variables in a normalizing flow model from the origin, assuming that the distribution defined over the latent space is an isotropic Gaussian. This measure only corresponds to measuring typicality if the bijection is volume preserving.

There are many other methods for performing anomaly detection which do not rely on traditional hypothesis testing theory. For example, the works in Hendrycks & Gimpel (2016); Hendrycks et al. (2018); Liang et al. (2017) propose using the outputs of trained neural networks themselves to discern when the networks are presented with out-of-distribution data. By examining the soft-max probabilities at the penultimate layer of these networks, they can distinguish between in- and out-of-distribution inputs with relatively good success. The work in Schlegl et al. (2017) similarly uses the output of the final layers in the Discriminator in a GAN to detect anomalous data-points. The authors in McAllister et al. (2019) use a VAE to make data points more likely under a learned distribution before feeding them to a trained model. An issue with all of these methods mentioned is that they rely on checking whether low-dimensional, learned representations of the data are anomalous or not. This has the downside that by construction, anomalous aspects of the data can be missed if the low-dimensional representation is invariant to those aspects. Nevertheless, some of these methods have proven to work well in certain domains. A survey of other methods for leveraging deep learning for anomaly detection can be found in in Chalapathy & Chawla (2019).

As an important aspect of hypothesis testing, it is important to consider the many methods, old and new, for learning probability densities. Kernel Density Estimation (KDE) methods are a traditional way of estimating densities, although they do not scale to high-dimensional large data sets. An overview of KDE methods can be found in Chen (2017). Some more modern approaches to modeling densities were defined in Section 2.3. In addition to the parameterizations of densities discussed there, many extensions have been developed with appealing properties, such as Chen et al. (2017), Ho et al. (2019), Grathwohl et al. (2018), De Cao et al. (2019), van den Oord et al. (2017), and Razavi et al. (2019), to list a few. There are also other objectives that are considered when learning densities, such as training a density in a adversarial manner as in Grover et al. (2018) and Danihelka et al. (2017). Alternatively, Li & Malik (2018) define implicit maximum likelihood estimation, which can be thought of as approximately minimizing the reverse KL-divergence.

# 6 CONCLUSION

We have presented a case study investigating why learned density models can perform poorly when used in goodness-of-fit tests. We argue that the hypothesis test used to test for anomalous data must be considered together with the procedure for learning a density which is used by the test. Along these lines, we propose training an ensemble of learned densities and jointly testing for typicality with respect to each of these densities. We proved that the intersection of typical sets of such an ensemble can lie in the interior of the typical set of the ground-truth data distribution if the ensemble is constructed correctly. We demonstrated on a simple and realistic example that in practice, this procedure outperforms using typicality tests on a single learned density.

Investigations left for future work include extending error rate bounds to the case of the multi-typicality test proposed here, and further evaluating the empirical performance of our proposed test on different domains.

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

## A  APPENDIX

### A.1  PROOF OF THEOREMS

To prove Theorem 1, we make use of the following lemma.

**Lemma 1.** *Consider a continuous random variable* $x \in \mathcal{X}$, *and two distributions* $q_a(x)$ *and* $q_b(x)$. *Denote the differential entropy of* $q_a(x)$ *and* $q_b(x)$ *as* $h_a$ *and* $h_b$, *respectively. Let* $D_{\mathrm{KL}}(q_a \| q_b) = d$. *Denote the typical sets of these distributions as* $T_a := T_\epsilon^{(n)}(q_a(x))$ *and* $T_b := T_\epsilon^{(n)}(q_b(x))$. *Finally, let* $T_{ab} := T_\epsilon^{(n)}(D_{\mathrm{KL}}(q_a \| q_b)$ *represent the relative entropy typical set (Cover & Thomas (2012), Section 11.8). Then, for* $n$ *sufficiently large,*

$$q_a(T_b) < e^{-n(d+h_a-h_b-3\epsilon)} + 3\epsilon.$$

*Proof of Lemma 1.*  We prove this lemma through contradiction. Assume

$$q_a(T_b) > e^{-n(d+h_a-h_b-3\epsilon)} + 3\epsilon.$$

Then by Theorem 8.2.2 in Cover & Thomas (2012) and the union bound, we have

$$q_a(T_a \cap T_b) > e^{-n(d+h_a-h_b-3\epsilon)} + 2\epsilon,$$

and, $q_a(T_a \cap T_{ab}) > 1 - 2\epsilon$. This implies that

$$e^{-n(d+h_a-h_b-3\epsilon)} < q_a(T_a \cap T_b \cap T_{ab})$$
$$< e^{-n(h_a-\epsilon)} Vol(T_a \cap T_b \cap T_{ab}),$$

implying

$$e^{-n(d-h_b-2\epsilon)} < Vol(T_a \cap T_b \cap T_{ab}).$$

We also have that

$$q_b(T_{ab}) > q_b(T_a \cap T_b \cap T_{ab}) > Vol(T_a \cap T_b \cap T_{ab}) \cdot \min_{x \in T_a \cap T_b \cap T_{ab}} q_b(x)$$
$$> e^{-n(d-h_b-2\epsilon)} e^{-n(h_b+\epsilon)}$$
$$= e^{-n(d-\epsilon)}$$

However, we also have that $q_b(T_{ab}) < e^{-n(d-\epsilon)}$ by Lemma 11.8.1 in Cover & Thomas (2012). This implies that

$$e^{-n(d-\epsilon)} < e^{-n(d-\epsilon)},$$

which forms a contradiction. Therefore, our original assumption must be false, and therefore

$$q_a(T_b) < e^{-n(d+h_a-h_b-3\epsilon)} + 3\epsilon.$$

□

*Proof of Theorem 1.*  The proof relies on the preceding lemma. Letting $T_b$ in Lemma 1 correspond to our acceptance region for $\mathcal{H}_0$, $T_\epsilon^{(n)}(p(x))$, then Lemma 1 states that for any distribution $\tilde{p}$, the probability

$$\tilde{p}(T_\epsilon^{(n)}(p(x))) < e^{-n(D_{\mathrm{KL}}(\tilde{p}\|p)+h_{\tilde{p}}-h_p-3\epsilon)} + 3\epsilon.$$

Therefore, by optimizing over all distributions $\tilde{p}(x)$ such that $D_{\mathrm{KL}}(p\|\tilde{p}) < d$, the result follows immediately.  □

To prove Theorem 2, we make use of the following lemmas.

**Lemma 2.** *For* $n$ *sufficiently large,*

$$q_k(T_p) > (1 - 2\epsilon)e^{-n(d_k+\epsilon)}$$

*Proof of Lemma 2.*  See Cover & Thomas (2012), Lemma 11.8.1.  □

**Lemma 3.** *For $n$ sufficiently large,*

$$Vol\Big(T_p \cap T_{q_k}\Big) \geq (1 - 2\epsilon)e^{n(h(p)-3\epsilon)} - \epsilon e^{n(h(p)+d_k-2\epsilon)}.$$

*Proof of Lemma 3.* For $n$ sufficiently large, $q_k(T_{q_k}) > 1 - \epsilon$, as shown in Cover & Thomas (2012), Theorem 8.2.2. By the union bound and Lemma 2,

$$q_k(T_{q_k}^{\mathbf{c}} \cup T_p^{\mathbf{c}}) < \epsilon + 1 - (1 - 2\epsilon)e^{-n(d_k+\epsilon)}$$

$$(1 - 2\epsilon)e^{-n(d+\epsilon)} - \epsilon < q_k(T_{q_k} \cap T_p)$$

$$\leq \int_{\mathbf{x} \in T_{q_k} \cap T_p} p(\mathbf{x}^n)e^{-n(d_k-\epsilon)}d\mathbf{x}^n$$

$$\leq \int_{\mathbf{x} \in T_{q_k} \cap T_p} e^{-n(h(p)-\epsilon)}e^{-n(d_k-\epsilon)}d\mathbf{x}^n$$

$$= e^{-n(h(p)+d_k-2\epsilon)}Vol\Big(T_{q_k} \cap T_p\Big)$$

$$(1 - 2\epsilon)e^{n(h(p)-3\epsilon)} - \epsilon e^{n(h(p)+d_k-2\epsilon)} < Vol\Big(T_{q_k} \cap T_p\Big)$$

$\square$

**Lemma 4.** *For sets $S_i \subset \mathcal{X}, i \in \{0, ..., K\}$, if*

$$\frac{Vol(S_i \cap S_0)}{Vol(S_0)} > \frac{K-1}{K}, \quad \forall i \in \{1, ..., K\},$$

*Then $Vol(\bigcap_{i=0}^{K} S_i) > 0$.*

*Proof of Lemma 4.* Immediate consequence of union bound. $\square$

*Proof of Theorem 2.* For $n$ sufficiently large, $Vol\big(T(p)\big) < e^{n(h(p)+\epsilon)}$ (Cover & Thomas (2012), Theorem 8.2.2). If for all $k \in \{1, ..., K\}$,

$$\frac{Vol(T(q_k) \cap T(p))}{Vol(Tp)} > \frac{K-1}{K},$$

then by Lemma 4, the result holds. For $k \in \{1, ..., K\}$,

$$\frac{Vol(T(q_k) \cap T(p))}{Vol(T(p))} > \frac{(1 - 2\epsilon)e^{n(h(p)-3\epsilon)} - \epsilon e^{n(h(p)+d_k-2\epsilon)}}{e^{n(h(p)+\epsilon)}}$$

$$= (1 - 2\epsilon)e^{-4n\epsilon} - \epsilon e^{n(d_k-3\epsilon)}.$$

Therefore if

$$\frac{K-1}{K} < (1 - 2\epsilon)e^{-4n\epsilon} - \epsilon e^{n(d_k-3\epsilon)},$$

or equivalently

$$d_k < \frac{1}{n}\log\Big(\frac{1 - 2\epsilon}{\epsilon}e^{-4n\epsilon} - \frac{K-1}{\epsilon K}\Big) + 3\epsilon,$$

then the result follows immediately. Note that the following condition is only valid if the argument of the logarithm is positive, and only possible if the entire right-hand side is positive. An sufficient condition for the RHS to be positive is,

$$\frac{1}{4\epsilon}\log\Big(\frac{1 - 2\epsilon}{\frac{K-1}{K} + \epsilon}\Big) > n.$$

These conditions are the two conditions in Theorem 2. $\square$

*Proof of Theorem 3.* Similar to Lemma 1, we prove this by contradiction. We make use of bounds that hold for $n$ sufficiently large, which we assume from here on to avoid repeatedly stating so. Let $T_{ab} := T_\epsilon^{(n)}(D_{\mathrm{KL}}(q_a \| q_b))$ represent the relative entropy typical set (Cover & Thomas (2012), Section 11.8).

Assume that, in fact,

$$\frac{Vol(T_a \cap T_b)}{Vol(T_a)} > r.$$

Therefore, by Theorem 8.2.2 in Cover & Thomas (2012),

$$Vol(T_a \cap T_b) > rVol(T_a)$$
$$> r(1 - \epsilon)e^{n(h_a - \epsilon)}.$$

However, by the definition of $T_a$,

$$Vol(T_a \cap T_b) < \frac{q_a(T_a \cap T_b)}{\min_{x^n \in T_a \cap T_b} q_a(x^n)}$$
$$< \frac{q_a(T_a \cap T_b)}{e^{-n(h_a + \epsilon)}}.$$

This then implies that

$$q_a(T_a \cap T_b) > r(1 - \epsilon)e^{-2n\epsilon}.$$

Now, using the union bound and Theorem 11.8.2 in Cover & Thomas (2012), we have that

$$q_a(T_a \cap T_{ab}) > 1 - 2\epsilon.$$

Therefore, again by the union bound,

$$r(1 - \epsilon)e^{-2n\epsilon} - 2\epsilon < q_a(T_a \cap T_b \cap T_{ab})$$
$$= \int_{x^n \in T_a \cap T_b \cap T_{ab}} q_a(x^n)dx^n$$
$$< e^{-n(h_a - \epsilon)}Vol(T_a \cap T_b \cap T_{ab})$$
$$\implies V(T_a \cap T_b \cap T_{ab}) > (r(1 - \epsilon)e^{-2n\epsilon} - 2\epsilon)e^{n(h_a - \epsilon)}.$$

Finally, we have that

$$q_b(T_{ab}) \geq q_b(T_a \cap T_b \cap T_{ab}) \geq Vol(T_a \cap T_b \cap T_{ab}) \cdot \min_{x^n \in T_a \cap T_b \cap T_{ab}} q_b(x^n)$$
$$= Vol(T_a \cap T_b \cap T_{ab})e^{-n(h_b + \epsilon)}$$
$$> (r(1 - \epsilon)e^{-2n\epsilon} - 2\epsilon)e^{-n(h_b - h_a + 2\epsilon)}.$$

From Theorem 11.8.2 referenced above, we also have that $q_b(T_{ab}) < e^{-n(D_{\mathrm{KL}}(q_a \| q_b) - \epsilon)}$, which implies the following.

$$(r(1 - \epsilon)e^{-2n\epsilon} - 2\epsilon)e^{-n(h_b - h_a + 2\epsilon)} < e^{-n(D_{\mathrm{KL}}(q_a \| q_b) + \epsilon)}$$
$$\log (r(1 - \epsilon)e^{-2n\epsilon} - 2\epsilon) - n(h_b - h_a + 2\epsilon) < -n(D_{\mathrm{KL}}(q_a \| q_b) - \epsilon).$$

Rearranging, this gives rise to the condition:

$$D_{\mathrm{KL}}(q_a \| q_b) > h_b - h_a - 3\epsilon + \frac{1}{n}\log \left( \frac{1}{r(1 - \epsilon)e^{-2n\epsilon} - 2\epsilon} \right).$$

If this condition does not hold, then there is a contradiction, and our original assumption that

$$\frac{Vol(T_a \cap T_b)}{Vol(T_a)} < r$$

must be false. $\qquad \square$

## A.2 Expressing the Multi-Typical Set as the Typical Set of some Distribution

Recall the definition of the multi-typical set for a given ensemble of probability distributions:

$$T_\epsilon^{(n)}(\{q_1(\mathbf{x}), ..., q_K(\mathbf{x})\}) := \left\{ \mathbf{x}^n \in \mathcal{X}^n : \max_{k \in \{1, ..., K\}} | -\log q_k(\mathbf{x}^n) - h_k | < \epsilon \right\}. \tag{8}$$

An interesting question to ask is if there exist a probability distribution which has typical set corresponding to the multi-typical set defined here. Here we show that we can approximately recover an un-normalized version of such a distribution.

Define $\tilde{q}_k(\mathbf{x}) := e^{h_k} q_k(\mathbf{x})$. Then we have

$$T_\epsilon^{(n)}(\{q_1(\mathbf{x}), ..., q_K(\mathbf{x})\}) = \left\{ \mathbf{x}^n \in \mathcal{X}^n : \max_{k \in \{1, ..., K\}} |\log \tilde{q}_k(\mathbf{x}^n)| < \epsilon \right\}. \tag{9}$$

.

Defining $\tilde{m}(x) := \tilde{q}_k(x),\ k = arg \min_i |\log \tilde{q}_i(x)|$, we have

$$T_\epsilon^{(n)}(\{q_1(\mathbf{x}), ..., q_K(\mathbf{x})\}) = \left\{ \mathbf{x}^n \in \mathcal{X}^n : |\log \tilde{m}(\mathbf{x}^n)| < \epsilon \right\}. \tag{10}$$

.

Now, define $Z = \int_{\mathbf{x} \in \mathcal{X}} \tilde{m}(\mathbf{x}) d\mathbf{x}$, and $m(\mathbf{x}) = \tilde{m}(\mathbf{x})/Z$ such that $m(\mathbf{x})$ is a valid probability distribution. Then if $\mathbb{E}_m[-\log m(x)] = \log(Z)$, then the distribution $m(x)$ has typical set identical to the set $T_\epsilon^{(n)}(\{q_1(\mathbf{x}), ..., q_K(\mathbf{x})\})$. The property that $\mathbb{E}_m[-\log m(\mathbf{x})] = \log(Z)$ will not hold exactly in practice, but often times it is approximately true. The reason this is true is because over the space $\mathcal{X}$, each of the functions $\log \tilde{q}_k(\mathbf{x})$ have zero mean. Pointwise, the function $\tilde{m}(x)$ takes as its value the $\tilde{q}_k$ whose log value is farthest from the origin. If the $\tilde{q}_k$ are chosen such that on average, equal mass is kept on either side of the origin (in log-space), then the property will hold.

The result of such a procedure is a approximation of the density which has typical set identical to the multi-typical set. Since in practice computing the normalization constant $Z$ is intractable, we can learn another divergence, using $\tilde{m}(\mathbf{x})$ as a target instead of $p(\mathbf{x})$. The reason we might wish to do this, is by having an explicit form of $\tilde{m}(\mathbf{x})$, we can learn a density by optimizing other divergences than the forward KL-divergence, such as the reverse KL-divergence. Note that if we optimize

$$D_{\text{KL}}(q(x) \| \tilde{m}(x)) = D_{\text{KL}}(q(x) \| Z m(x)) \tag{11}$$
$$= D_{\text{KL}}(q(x) \| m(x)) - \log Z. \tag{12}$$

Therefore, knowledge of the normalization constant is not needed in this context, since the constant term doesn't affect the optimization procedure. Similar results can be shown for other divergences.

## A.3 Visualizations of Samples from the Mixture of Gaussians Experiment

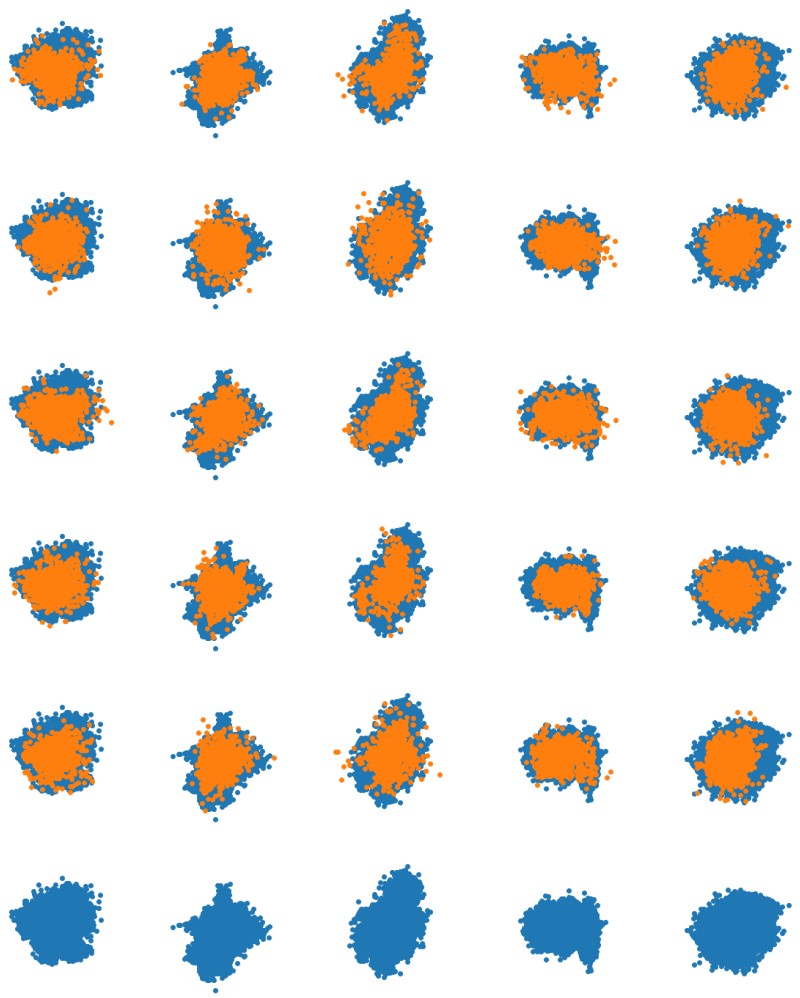

Figure 5: Samples from learned densities which are parameterized as Mixture of Gaussians. Random projections (columns) of samples from different learned densities (rows), are overlaid on samples from the target Mixture of Gaussians distribution. The bottom row displays the ground-truth samples unobstructed for reference.

