# OpenReview forum: "Testing For Typicality with Respect to an Ensemble of Learned Distributions"
_ICLR.cc/2020/Conference — Reject_

### Official Review · AnonReviewer1 · 2019-10-14
**Official Blind Review #1**

**Rating:** 1

**Review:**

Summary:  This paper analyzes and extends a recently proposed goodness-of-fit test based on typicality [Nalisnick et al., ArXiv 2019].  Firstly, the authors give bounds on the type-II error of this test, showing it can be characterized as a function of KLD[q || p_true] where p is the true data generating process and q is an alternative generative process.  The paper then shifts to the main contribution: an in-depth study of a Gaussian mixture simulation along with accompanying theoretical results.  The simulation shows that maximum likelihood estimation (MLE)---due to it optimizing KLD[p_true || p_model]---does not penalize the model for placing probability in places not occupied by p_true.  This means that while samples from p_true should fall within the model’s typical set, the model typical set may be broader than p_true’s.  Table 1 makes this clear by showing that only 30-40% of samples from the model fall within the typical set of p_true.  Yet >93% of samples from p_true fall within the models’ typical sets.  The paper then makes the observation that the models do not have high overlap in their typical sets, and thus p_true’s typical set could be well approximated by the intersection of the various models’ typical sets.  Applying this procedure to the Gaussian mixture simulation, the authors observe that ~95% of samples drawn from the intersection of the ensemble fall within p_true’s typical set.  Moreover, ~97% of samples from p_true are in the ensemble (intersection) typical set.  The paper closes by proving that the diversity of the ensemble controls the overlap in their typical sets, and hence increasing diversity should only improve the approximation of p_true’s typical set.

____

Pros:  This paper contributes some interesting ideas to a recent topic of interest in the community---namely, that deep generative models assign high likelihood to out-of-distribution (OOD) data [Nalisnick et al., ICLR 2019] and how should we address this problem if we are to use them for anomaly detection, model validation [Bishop, 1994], etc.  This paper makes some careful distinctions between the true data process, the model, and the alternative distribution, which I have not seen done often in this literature.  And while the mass-covering effect of MLE on the resulting model fit is well known, this paper is the first with which I am aware that translates that fact into a practical recommendation (i.e. their intersection method).  Furthermore, this connection to ensembling may provide important theoretical grounding to other ensemble-based methods for OOD detection [Choi et al., ArXiv 2019].

____

Cons:  The primary deficiency in the paper is experimental.  While the text does make some compelling arguments in the Gaussian mixture simulations, some validation on real data must be provided.  Ideally experiments on CIFAR-10 vs SVHN (OOD) and FashionMNIST vs MNIST (OOD) should be reported as these data set pairings have become the benchmark cases in this line of literature.

Besides the lack of experiments on real data, I find the paper’s material to be a bit disjointed and ununified.  For instance, Theorem 1 is never discussed again after it is presented in Section 2.1.  I thought for sure the presence of the KLD-term would be referenced again to relate the ensembling methodology back to the bound on the type-II error.  For another example, normalizing flows are discussed in Section 2.3 and the change-of-variables formula given in Equation 5.  However, normalizing flows are never mentioned again except in passing in the Related Work section.

____

Final Evaluation:  While I find the paper to contain interesting ideas, it is too unfinished for me to recommend acceptance at this time.  Experiments on real data must be included and the overall coherence of the draft improved.


**Experience Assessment:**

I have published one or two papers in this area.

**Review Assessment: Checking Correctness Of Derivations And Theory:**

I assessed the sensibility of the derivations and theory.

**Review Assessment: Checking Correctness Of Experiments:**

I assessed the sensibility of the experiments.

**Review Assessment: Thoroughness In Paper Reading:**

I read the paper thoroughly.

---

> ### Comment · AnonReviewer1 · 2019-11-14
> **Post-Reviews Update**
>
> After reading the two other reviews, it seems that all reviewers agree that the lack of non-toy experiments is a major deficiency in the paper.  I find R3 to be too harsh in claiming Eqs 3 and 4 are "wrong": the authors clearly show $p$ is approximated with samples in Eq 4.  I also disagree with R3 that the included experiment is "unclear".  Rather, I find its motivation and results easy to interpret (see my summary).  I mostly concur with R2's review except for its final conclusion.  The questions that R2 lists under 'cons' could indeed be answered with a more comprehensive and realistic set of experiments.  And until high-dimensional experiments are included, I leave my recommendation at "reject".

---

### Official Review · AnonReviewer3 · 2019-10-15
**Official Blind Review #3**

**Rating:** 3

**Review:**

This paper proposes to use ensembles of estimated probability distributions in hypothesis testing for anomaly detection.
While the problem of density estimation with its application to anomaly detection is relevant, I have a number of concerns listed below:

- Overall, this paper is not clearly written and it is difficult to follow.
    Discussion is not straightforward at many points.
    In particular, the objective of experiments on synthetic data in Section 3 is unclear. What is the proposal and how to evaluate it in the experiments?
    There are also many grammatical mistakes, which also deteriorates the quality of the paper.
- Technical quality is not high.
    * Equations (3) and (4) are wrong. The distribution p should be not the ground truth but the empirical distribution.
    * In experiments, only a simple Gaussian mixture model has been examined. A variety of distributions should be examined.
    * How strong are the assumptions in Theorem 2 in practical situations?
- There is no experimental evaluation for the proposed method. Hence the effectiveness of the proposed method is not clear.

Minor comments:
- P.2, L.3 in Section 2: ", The" -> ", the"
- P.7, L.-6: "q_1(x; \theta_1," -> "q_1(x; \theta_1),"
- P.8, L.1 in Theorem 2: "x \in X, Consider" -> "x \in X, consider"


**Experience Assessment:**

I have published one or two papers in this area.

**Review Assessment: Checking Correctness Of Derivations And Theory:**

I assessed the sensibility of the derivations and theory.

**Review Assessment: Checking Correctness Of Experiments:**

I carefully checked the experiments.

**Review Assessment: Thoroughness In Paper Reading:**

I read the paper at least twice and used my best judgement in assessing the paper.

---

> ### Comment · AnonReviewer3 · 2019-11-14
> **Post-Reviews Update**
>
> I have read the other two reviews and agree that the major deficiency is the lack of non-toy experiments.
> I also agree with most of the points raised by R1 and R2.
> Eq.(3) becomes correct if p is an empirical distribution, while in the paper p is referred to as the ground truth distribution.
> Since it usually does not mean the empirical distribution, this point should be clarified.

---

### Official Review · AnonReviewer2 · 2019-10-22
**Official Blind Review #2**

**Rating:** 6

**Review:**

Summary:

I machine learning, we often have training data representative of an underlying distribution, and we want to test whether additional data come from the same distribution as the training data (e.g. for outlier/anomaly detection, or model checking). One way to do this is to learn a model of the underlying distribution, and test whether the additional data fall within the typical set of the model. This paper points out that the typical set of the model may be very different from the typical set of the underlying distribution if the model is learned by maximum likelihood, in which case a test of typicality with respect to the model would be a poor test of typicality with respect to the underlying distribution. The paper shows theoretically that the intersection of the typical sets of an ensemble of models lies within the typical set of the underlying distribution, provided that (a) each model is a good enough approximation to the underlying distribution, and (b) the models are all sufficiently different from each other. Based on that, the paper argues that a better test of typicality would be to test whether the additional data fall within the intersection of the typical sets of the ensemble of models.

Pros:

The paper addresses an interesting problem in a sound and well motivated way. There is a lot of work on outlier/anomaly detection that uses the model's probability density to determine whether a dataset is out-of-distribution or not, which is known to not be a good proxy for typicality, because atypical data can have high probability density. In contrast, this paper uses a well-founded notion of typicality based on the information-theoretic definition of a typical set.

The toy example that is used to illustrate the problem is clear and illuminating, and motivates the paper well. In particular, the example clearly illustrates the issue of local minima when training models, and the mass-covering behaviour of maximum-likelihood training.

The idea of using the intersection of the typical sets of an ensemble of models is interesting and clever, and backed by strong theoretical results.

Cons:

Even though I appreciate the paper's theoretical contribution, there are no empirical results other than the motivating example. In particular, the paper proposes an idea and theory to back it up, but it doesn't really propose a practical method, and as a result it doesn't test the theory in practice.

Theorems 2 and 3 provide a solid foundation for the proposed idea, but it's not clear how they can be used in practice. Specifically:
- How can we verify that in practice the KL between the models and the underlying distribution is small enough as required by theorem 2 when we can't usually evaluate it?
- In practice, how should we construct an ensemble such that the individual models in the ensemble are different enough from each other as required by theorem 3?
- Both theorem 2 and 3 are valid "for large enough n". However, in practice we may want to check e.g. individual datapoints for typicality (in which case n=1). Are the theorems relevant for small n?

The paper is generally well written, but some statements made are either inaccurate or subjective, and I worry that they might mislead readers. Later in my review I will point out exactly which statements I'm referring to. I strongly encourage the authors to fix or moderate these statements before the paper is published.

Decision:

I believe the paper to be an important contribution, but the work is clearly incomplete. For this reason, my recommendation is weak accept, with an encouragement to the authors to continue the good work.

Inaccuracies or subjective statements that I encourage the authors to fix/moderate:

"we are still bad at reliably predicting when those models will fail"
"we are unable to detect when the models are presented with out-of-distribution data"
These statements may come across as too strong. I suggest making the statements about our current methods, rather than about the ability of the research community, and be more specific in what ways the current methods are inadequate.

"detecting out-of-distribution data [...] is formally known as the goodness-of-fit problem"
I'm not sure that detecting our-of-distribution data and goodness-of-fit are synonymous. Goodness-of-fit testing can be used in situations other than outlier detection, e.g. for testing whether a proposed model is a good fit to a dataset.

(Second bullet-point of section 1) "distributions having low KL divergence must have non-zero intersection"
To be more precise, the typical sets must have non-zero intersection, not the distributions.

"determining which of two hypotheses are more probable"
"H0 is deemed more probable"
Classical hypothesis testing does not assign a probability to a hypothesis, which would be a Bayesian approach instead. Therefore, it's technically incorrect to talk about the probability of a hypothesis in this context.

"which accepts the null-hypothesis"
"f correctly accepting H0"
Hypothesis testing doesn't accept a hypothesis, it merely decides whether to reject the null hypothesis in favour of the alternative hypothesis. Therefore, it may "fail to reject" the null hypothesis, but it never accepts it.

"the KL-divergence is equal to zero if and only if p(x) = q(x; θ) ∀x ∈ X"
The KL is equal to zero if and only if the distributions are equal, but the densities may still differ in at most a set of measure zero. Therefore, it's not a requirement that the densities match for all x for the KL to be zero.

"For example, by looking at the form of the KL-divergence, there is no direct penalty for q(x; θ) in assigning a high density to points far away from any of the ¯xi’s"
The problem that this statement is talking about is the problem of overfitting, which is the problem of the model learning the specifics of the training data rather than the underlying distribution. However, the statement preceding the above is about the problem of local minima when optimizing then parameters of a model. These two problems are distinct and shouldn't be conflated, as they are here.

"this requires direct knowledge of p(x) to evaluate the objective"
However we can evaluate the objective up to an additive constant when p(x) is known up to a multiplicative constant, which is enough to optimize it.

"as do all divergences other than the forward KL, to the best of our knowledge"
"This makes the forward KL-divergence special in that it is the only divergence which can directly be optimized for."
I don't think this is true. For example, the Maximum Mean Discrepancy is a divergence, since it's non-negative and zero if and only if the two distributions are equal, but it only involves expectations under p(x) and can be directly optimized over the parameters of q(x; \theta). Moreover, the second statement doesn't follow from the first: it's incorrect to conclude that the forward KL is the only one that can be directly optimized for, based only on one's state of knowledge.

"Variational Auto-encoders [...] map a lower-dimensional, latent random variable"
There is no fundamental reason why the latent variable of a VAE has to be low-dimensional. We may do this often in practice, but a VAE with a high-dimensional latent variable may also be used.

"Because the image of any non-surjective function necessarily has measure zero"
This is not true; the absolute-value function is not surjective but its image doesn't have measure zero in the set of real numbers. I understand what the statement is trying to say, but it's important that it's said accurately.

"autoregressive models, such as PixelCNN"
Autoregressive models can also be used to model discrete variables in which case they can't be thought of as flows. In fact, PixelCNN as first proposed is a model of discrete variables.

"all of these models rely on optimizing the forward KL-divergence in order to learn their parameters"
Not necessarily, flow-based models don't have to be optimized by minimizing the forward KL. For example, they can be trained adversarially in the same way as GANs, and in principle can be trained with other divergences or integral probability metrics. The model and the loss are (at least in principle) orthogonal choices.

"advancements in the expressivity of the models are unlikely to fix the undesired effects"
This is a subjective assessment, and is not sufficiently backed by arguments where it first appears. I understand that the arguments are presented later in section 3, so I would at least suggest that a forward reference to the argumentation in section 3 is given here.

Figure 5 gives the impression that the model samples have less variance than the ground-truth samples. Isn't that surprising given that the problem is that minimizing the forward KL leads to mass-covering behaviour? I suspect that the problem here is that there are more ground-truth samples than model samples, and the ground-truth samples saturate the scatter plot. If that's the case, I believe that figure 5 is very misleading.

"we see that the learning procedure converged"
We know however that the learning procedure hasn't really converged, instead it is stuck at a saddle point (where the model is using a single mode to cover two modes of the underlying distribution). In other words, it appears to us that the learning procedure has converged, even though it hasn't, and possibly if we wait for long enough we will see rapid improvement when the procedure escapes the saddle point. Therefore, I would at least say "we see that the learning procedure has appeared to converge".

I would expect the bottom-right entry of table 1 to be higher than 90% like the other diagonal elements, so I suspect that it might be a typo.

In eq. (7), shouldn't each log q_k be divided by n?

"in practice we find that it is much easier to find an ensemble of models such that the multi-typical set approximates the ground-truth typical set than the bounds require"
There is no empirical evidence presented in the paper in support of this statement.

"least probable density"
"least typical density"
I understand what the intended meaning of these terms is, but these terms make little sense mathematically nevertheless. I would suggest that the statement is rewritten in a more precise and direct way.

"This measure only corresponds to measuring typicality if the bijection is volume preserving"
I'm not sure that the distance from a Gaussian mean is a valid measure of typicality. In high dimensions, the region around the mean is very atypical.

Minor errors, typos, and suggestions for improvement:

The phrase "the authors in Smith et al. (2019) propose" is a bit awkward. Better say "Smith et al. (2019) propose", as Smith et al are indeed the authors.

Missing full stop in first bullet-point of section 1.

It would be good to provide more details of the experiment in section 3. Specifically:
- What training algorithm was used to maximize the likelihood? SGD or EM?
- How many training datapoints were used?

"to index the 5 experiments ran" --> run

"refer the k-th learned density" --> refer to

interestig --> interesting

Missing closing bracket in point 1 of section 4.

Capital C in "Consider" in theorem 2.

"if every model in a density of learned distributions" --> an ensemble of learned distributions

"where as the method we propose" --> whereas

"can be found in in", double "in"

**Experience Assessment:**

I have published one or two papers in this area.

**Review Assessment: Checking Correctness Of Derivations And Theory:**

I assessed the sensibility of the derivations and theory.

**Review Assessment: Checking Correctness Of Experiments:**

I carefully checked the experiments.

**Review Assessment: Thoroughness In Paper Reading:**

I read the paper thoroughly.

---

### Public Comment · ~Shengyu_Zhu1 · 2019-10-17
**Interesting. Yet questionable motivation and example, as well as missing definition/detail**

It is really interesting to see that typicality, which I used quite a lot in my previous research, is also considered with machine learning. I have several questions with the current manuscript.

1. Motivation: in your abstract and introduction, you mentioned

'which is known as the goodness-of-fit problem, but those results require knowing a model of the base distribution. The ability to correctly reject anomalous data hinges on the accuracy of the model of the base distribution. For high dimensional data, learning an accurate-enough model of the base distribution such that anomaly detection works reliably is very challenging, as many researchers have noted in recent years. Existing methods for the goodness-of-fit problem do not account for the fact that a model of the base distribution is learned. '.

'These type of bounds show that certain tests can be performed which are capable of discerning (with non-trivial probability) that populations of data sampled from distributions at least some positive distance away from the base distribution are anomalous. However, in order to perform the proposed tests, an explicit form of the probability density function (or probability mass function) describing the base distribution is needed. For most real-world data sets, this density is not known, and must be estimated. While there has been a lot of analysis on the ability to detect anomalous data, those analyses typically do not account for the fact that the base density for which the tests are designed is learned'

This is questionable. In the goodness of fit setting (also called one-sample problem), it is true that one has to consider the base distribution because this is the problem setting: a distribution is given and one tries to test how well this distribution fits observed data. However, if this distribution is not given but instead one has samples, it is straightforward to conduct a two-sample testing (e.g., using MMD). The current motivation of estimating the base distribution is not convincing.

2. Theorem 1: in my experience with information theory (particularly with source coding and hypothesis testing), the result of this theorem cannot be called 'error rate', since you have a constant term $3\epsilon$ on the r.h.s. In other words, it is not clear how you pick this $\epsilon$ and this result does not indicate consistency (consistency: the type-II error probability goes to zero with $n\to\infty$, subject to a fixed type-I error probability). Indeed, many nonparametric goodness of fit tests are consistent, and using entropy typicality set of $p$ as acceptance region is far from optimal testing in either finite or asymptotic regime wrt. error probabilities. Actually a recent work has also shown the MMD based test, applied to goodness of fit testing, is universally optimal in the sense that it achieves the optimal type-II error exponent for any alternative distribution with $\text{KLD}<\infty$, with a fixed type-I error probability. Check https://arxiv.org/abs/1908.10037 if you are interested.

3. The mixture of Gaussian example is also questionable. It is not clear how you optimize the parameters (I didn't find details on this part). Using EM method could lead to a much better estimate. By the way, picking the right parametric model is usually not easy in practice.

4. Definition of multi-typicality in Eq. (7): I guess you mean $\min$, rather than $\max$; otherwise, as long as you have more than one $q_i$, you can find a small enough $\epsilon>0$ so that this set has nearly zero probability with sufficiently many samples from distribution $p$.

5. First line on Page 2: you assume $D_{\text{KL}}(p\|\tilde{p})\geq d$. What is $d$ and why this condition is placed? This isn't explained.

6. Minor:
- No definition of $Vol()$; this concept may be new to people in machine learning community
- No definition of 'intersection', so it is hard to verify your related claim
- Missing statement on the i.i.d. assumption on samples.
- Eq.(2) and Theorem 1: you use $\max$, but I don’t' think it is easy to see that the maximum indeed exists. So perhaps use $\sup$?

---

> ### Author Response · Authors · 2019-10-24
> **Thank you very much for your helpful comments**
>
> Dear Dr. Zhu,
>
> Thank you so much for your comments, and especially for linking your recent paper on the asymptotically optimal one- and two-sample kernel-based tests. In a final version of our paper we will be sure to reference your work, as it is very interesting and relevant. I will try to address all of your comments here, and will definitely address them in the final version of our paper, since they are all valid points.
>
> 1. You are correct that the two-sample problem directly addresses the problem of discerning distributions when only samples from the base distribution are given. The angle that we approached our work from was in response to recent proposals in the deep learning community that learning a succinct representation of the base distribution from samples to then use in one-sample testing could be advantageous over two-sample testing for computational reasons. I think that such a proposal is not made explicit in those works (nor as you point out, in our own), perhaps due to the known computational issues with sample-to-sample comparisons, e.g. nearest-neighbors.  I agree that we made a mistake in not mentioning two-sample testing at all, or the computational motivation in the test we propose. That being said, I do still think that the test we propose is still well-motivated, since for applications with massive, high-dimensional datasets, computing MMD- or KSD-based tests could be prohibitively slow to compute, where as the test we propose would not.  We will be sure to make this point very explicit in an updated version of the paper.
>
> 2. Again, you are correct that "error-rate" might not be the most appropriate term. Instead "probability of error" is a more accurate way to describe the term. Either way, while there is an additive $3\epsilon$ term on the bound, and a stronger bound would not include this additive term, the bound is still valid and gives some insight into what we can say about the power of the test on typicality proposed in (Nalisnick, 2019). I am curious about what you mentioned regarding tests based on the entropy typical set being far from optimal. Would you mind sharing a reference that includes that negative result? I would be very interested in reading about that.
>
> 3.  The mixture-of-gaussians example was chosen since it is the simplest example we could think of that demonstrated the phenomenon we were interested in showing. The same phenomenon can be seen for much more complicated examples, although in all examples the true base distribution must be known. This makes such an effect difficult to show on real-world datasets, and therefore examples can quickly become seemingly contrived. We thought that for clarity we would show that such effects are evident in one of the simplest of cases, but we understand that there are also limitations to focusing on such type of examples. Your comment is helpful, and in the final revision of our work we can instead/additionally demonstrate the effect on more complicated examples that are more similar to real-world datasets one might encounter in practice.
>
> Your comment regarding the optimization of parameters is also valid. In that example we optimized parameters using a gradient-based method with momentum (Adam) as is commonly used when optimizing the parameters of large flow-based generative models, and when the structure of the base-distribution is not known a priori.
>
> 4.  In the definition of multi-typicality, we do mean $\max$. This results in an acceptance region which is the intersection of the typical sets of each member in the ensemble. Theorem 2 gives sufficient conditions for such an intersection of typical sets to also have non-zero intersection with the typical set of the base distribution. These conditions do not guarantee that the resulting multi-typical set has large probability with respect to the base distribution, but provide a means to under-approximate the acceptance region defined by the test on entropy typicality ( as opposed to over-approximate the acceptance region, which is the usual result of using a learned approximation of the base typical set).
>
> 5.  This condition is to indicate that it may be impossible to define tests with non-trivial power for differentiating from the null hypothesis if alternate distributions can be assumed to be arbitrarily close to the base distribution. Admittedly, as you point out, we did not give this detail adequate consideration in the submitted version, and will address it in the revised version.
>
> 6. All noted and good points, will fix in revised version.
>
> Again, thank you very much for taking the time to read our submission and making thoughtful and informed comments. Each of your points are well founded and we look forward to incorporating them, and in doing so, strengthening our work.
>
> Kindly,
> The authors of submission 1992

---

> > ### Public Comment · ~Shengyu_Zhu1 · 2019-10-27
> > **Thanks for response**
> >
> > Dear authors,
> >
> > Thanks very much for your detailed response. I'm glad to see that the previous questions are indeed helpful and I think most of them can be handled by a little effort of revision. BTW, I also guessed that the reason of not using two-sample testing was due to computation issues, which in my opinion shall be mentioned to make the motivation more convincing.
> >
> > More about 'using entropy typical set of $p$ is far from optimal testing w.r.t. error probabilities':
> >
> > This is about the problem of goodness of fit testing in the universal setting. Given a distribution $p$ and i.i.d. samples denoted by $x^n$, decide whether or not $x^n$ are from $p$. Assume that $q$ is the true yet unknown distribution of $x^n$, then this problem can be formulated as $H_0: p=q$ vs. $H_1: p\neq q$. As the way you also used in the paper, for a fixed type-I error constraint, the question is: can we achieve the optimal type-II error probability for any $q$ even if we do not know $q$? For finite samples, I don't think it possible. There have been several (or many) works on the asymptotic case, that is, can we achieve the same type-II error exponent for any $q\neq p$?
> >
> > This problem probably dates back to W. Hoeffding in 1965 [1], where he showed the empirical KLD is indeed universally optimal for finite sample spaces. For more general space, like $\mathbb R$, only some weaker optimality results exist. In fact, I couldn not  find any reference for that statement about 'typical set'; I tried this entropy set to show achievability (existence) before but it was very hard (or impossible) to pick a universal $\epsilon$. Our work [2] actually solved this problem for at least $\mathbb R^n$. Please find more details and the above mentioned works in [2].
> >
> > BTW, this setting does not assume $\text{KLD} \geq d$. It basically assumes that $q\neq p$ (and in fact $KLD<\infty$ for regularization reasons). That said, as long as $q\neq p$, then no matter how close they are, they can be identified with sufficiently many i.i.d. samples. If assuming $\text{KLD}\geq d$, then one have constructed minimax optimal tests (sorry, I could not remember a reference).
> >
> > [1] Hoeffding, W. (1965). Asymptotically optimal tests for multinomial distributions. The Annals of Mathematical Statistics, 369-401.
> > [2] Asymptotically Optimal One- and Two-Sample Testing with Kernels. https://arxiv.org/abs/1908.10037

---

### Author Response · Authors · 2019-11-15
**Response to the reviewers**

We will address the response from all of you in a single comment here to avoid redundancy. First and foremost, thanks to each of you for taking the time to read the paper and offer your thoughts on our work.

It is clear that the main concern of all of the reviewers is that the paper lacks empirical results. We acknowledge that this is a fair criticism. However, we wish to defend a little our choice of this example, and respond to some of the proposals for other experiments. First, to evaluate the effectiveness of our proposed method, knowing the pdf of the base distribution is necessary to evaluate how well the learned distributions approximate the base. This makes evaluating on standard image datasets difficult. Being constrained to datasets in which the pdf of the base is known, we decided to stick with the most simple example in which the phenomenon we describe occurs. We made this choice for primarily three reasons: clarity, space, and computational resources. We thought it unnecessary to make the base distribution excessively complicated for fears that it would make the result seem contrived and potentially an artifact of the choice of base distribution.

R3 suggested that high-dimensional examples are still needed, but we argue that the 100 dimensional example shown is at least not low-dimensional. We felt that this example was sufficiently high dimensional to exhibit the effects of high-dimensional distributions, but low enough as to not require excessive computation and a finicky learning procedure.

Also on this note, R1 suggested that we make a dataset-to-dataset comparison as is done in other anomaly detection works. We believe that such comparisons are in a sense tangential to this work. While the comparisons made in those works do demonstrate the ability for the evaluated methods to reject samples from other chosen distributions, such a demonstration does not give confidence that the method will reject samples from some other distribution that was not evaluated against. In other words, such a comparison does not show how a model will perform on unknown unknowns, as a theoretical argument must be made for such a claim. Because prior work has already evaluated the effectiveness of using the typical set of a distribution as an acceptance region for one-sample tests, we felt that the most valuable use of space in our paper for experiments should go towards demonstrating the effectiveness of our method in better approximating the typical set itself.

Finally, we would also like to acknowledge the many other comments made by all of the reviewers about other or more minor things, which have all been received and for the most part we agree with. However, there are a few things we would like to respond to.

First, equations 3 and 4 are correct. Equation 3 is simply an objective. The objective that we wish to minimize can be anything we want -- whether we can evaluate and optimize for it is another question. The fact that in practice what is actually minimized is the KL between the empirical distribution of p and the parameterized distribution q is one interpretation. Another equally valid interpretation is that the objective that is  evaluate in practice is a sample approximation of the objective listed in (3), which is the view we take. We agree with R1 that to say this view is incorrect is itself an incorrect statement.

Finally, there was some questions about the application of the theoretical results to practical use. For example, R1 wondered why the result of theorem 1 never showed up again. It more or less does show up in the form of theorem 3, which states that for an ensemble of distributions which are sufficiently different will have low intersection. The metric that is used to show "difference" in distributions in this case is the same as that given in theorem 1.

The direct applicability of the theorems to practical situations, given as they are in a sense asymptotic results, is a fair thing to question, as R2 did. While we wish we were able to come up with strong bounds that are valid for any size sample, such analysis is very hard and might even be impossible without making further assumptions. This is left for future work. Instead, we just offer theorem 1, for example, as a means to give motivation to the test for typicality, which otherwise had no theoretical motivation that we are aware of, and theorems 2 and 3 as motivation for the idea of taking intersections of typical sets.

Noting all of that, we wish to again thank all of the reviewers for taking the time to think about the results we present. We are thankful for the criticisms as they are important for elevating our work.

---

> ### Comment · AnonReviewer1 · 2019-11-15
> **Re Author Response**
>
> Thank you for your response, authors.  And thanks for providing insight into your experimental decisions.  I agree that the Gaussian mixture simulation targets the issue directly.  I like this experiment---as I hope I made clear in my review---and think it should be in the paper.  However, I don't exactly see the tension between that simulation and a real-data experiment that your rebuttal seems to presume.  As I'm sure you know, papers often have simulations performed under ideal conditions and then experiments where those conditions might be violated.  Ultimately, there should be some practical consequence to the paper (in most cases, including this one).  I wish you the best of luck on your revisions.

---

### Decision · Program_Chairs · 2019-12-19

**Decision:**

Reject

**Comment:**

The paper proposes a new method for testing whether new data comes from the same distribution as training data without having an a-priori density model of the training data. This is done by looking at the intersection of typical sets of an ensemble of learned models.

On the theoretical side, the paper was received positively by all reviewers. The theoretical results were deemed strong, and the ideas in the paper were considered novel. The problem setting was considered relevant, and seen as a good proposal to deal with the shortcoming of models on out of distribution data.

However, the lack of empirical results on at least somewhat realistic datasets (e.g. MNIST) was commented on by all reviewers. The authors only present a toy experiment. The authors have explained their decision, but I agree with R1 that it would be appropriate in such situations to present the toy experiment next to a more realistic dataset. This also means that the effectiveness of the proposed method in real settings is as of yet unclear. Although the provided toy example was considered clear and illuminating, the clarity of the text could still be improved.

Although the reviewers had a spread in their final score, I think they would all agree that the direction this paper takes is very exciting, but that the current version of the paper is somewhat premature. Thus, unfortunately, I have to recommend rejection at this point.